# Labeling of Bruton’s Tyrosine Kinase (BTK) Inhibitor [^11^C]BIO-2008846 in Three Different Positions and Measurement in NHP Using PET

**DOI:** 10.3390/ijms25147870

**Published:** 2024-07-18

**Authors:** Sangram Nag, Prodip Datta, Anton Forsberg Morén, Yasir Khani, Laurent Martarello, Maciej Kaliszczak, Christer Halldin

**Affiliations:** 1Department of Clinical Neuroscience, Centre for Psychiatry Research, Karolinska Institutet and Stockholm County Council, SE-171 76 Stockholm, Sweden; sangram.nag@ki.se (S.N.); prodip.datta@ki.se (P.D.); anton.forsberg.moren@ki.se (A.F.M.); yasir.khani@ki.se (Y.K.); 2Biogen MA Inc., 225 Binney St., Cambridge, MA 02142, USA; lmartarello@sentonixpharma.com (L.M.); maciej.kaliszczak@biogen.com (M.K.)

**Keywords:** Bruton’s Tyrosine Kinase, Radiolabelling, PET, NHP

## Abstract

Bruton’s tyrosine kinase (BTK) is pivotal in B-cell signaling and a target for potential anti-cancer and immunological disorder therapies. Improved selective reversible BTK inhibitors are in demand due to the absence of direct BTK engagement measurement tools. Promisingly, PET imaging can non-invasively evaluate BTK expression. In this study, radiolabeled BIO-2008846 ([^11^C]BIO-2008846-A), a BTK inhibitor, was used for PET imaging in NHPs to track brain biodistribution. Radiolabeling BIO-2008846 with carbon-11, alongside four PET scans on two NHPs each, showed a homogeneous distribution of [^11^C]BIO-2008846-A in NHP brains. Brain uptake ranged from 1.8% ID at baseline to a maximum of 3.2% post-pretreatment. The study found no significant decrease in regional VT values post-dose, implying minimal specific binding of [^11^C]BIO-2008846-A compared to free and non-specific components in the brain. Radiometabolite analysis revealed polar metabolites with 10% unchanged radioligand after 30 min. The research highlighted strong brain uptake despite minor distribution variability, confirming passive diffusion kinetics dominated by free and non-specific binding.

## 1. Introduction

Bruton’s tyrosine kinase (BTK) is a key component in the B-cell receptor (BCR) signaling pathway, which plays a vital role in the development and activation of B cells [1,2]. In recent years, BTK inhibitors have attracted considerable attention as potential therapeutic agents for a range of diseases, including B-cell malignancies and autoimmune disorders [3]. An emerging and significant application of BTK inhibitors is in the field of cancer therapeutics. B-cell malignancies, such as B-cell lymphomas and chronic lymphocytic leukemia (CLL), often rely on BCR signaling for survival and proliferation [2,4,5]. BTK inhibitors have the ability to disrupt this signaling pathway, leading to the suppression of tumor growth and potential improvement in clinical outcomes for patients [6].

Furthermore, BTK inhibitors have demonstrated efficacy in the treatment of immunological disorders like rheumatoid arthritis [7,8] and multiple sclerosis (MS) [9]. In these conditions, aberrant BCR signaling contributes to the dysregulation of immune responses, leading to tissue damage [10]. By inhibiting BTK, these inhibitors can modulate the immune system and alleviate disease symptoms. To advance the development and use of BTK inhibitors, it is important to evaluate their target engagement and expression in vivo. Positron emission tomography (PET) imaging presents a noninvasive method to assess BTK levels and engagement with high specificity and sensitivity [11]. PET imaging relies on the use of radioligands, which are molecules labeled with a positron-emitting radioisotope. These radioligands can selectively bind to specific targets, allowing for the visualization and quantification of target expression and engagement in living organisms [12]. PET imaging of BTK can provide valuable information in several aspects [13,14]. First, it can aid in the selection of patients who are most likely to benefit from BTK inhibitor therapies. By identifying patients with high BTK expression, clinicians can tailor treatment strategies and determine the optimal dosage of BTK inhibitors. Second, PET imaging can help monitor treatment response and assess the effectiveness of BTK inhibitors. By quantifying BTK occupancy in target tissues, clinicians can evaluate if the inhibitor is effectively engaging the BTK receptors and adjust treatment plans accordingly. Additionally, BTK PET imaging offers the potential for noninvasive evaluation of treatment resistance. Monitoring changes in BTK expression and occupancy over time may help identify emerging resistance mechanisms and guide the development of strategies to overcome them. While BTK PET imaging holds great promise, there are still technical and logistical challenges that need to be addressed. The development of novel PET radioligands with improved pharmacokinetic properties and higher binding affinity are crucial for enhancing the sensitivity and specificity of BTK imaging.

In recent years, PET tracers targeting BTK, such as ^11^C-labeled analogs of tolebrutinib [15], ibrutinib [16], and evobrutinib [17], have been developed. Several ^18^F-labeled PET tracers targeting BTK, such as BTK-1 [18], [^18^F]ABBV-BTK1 [19], and PTBTK3 [20], have also been developed and evaluated for their potential utility preclinically. To the best of our knowledge, none of the aforementioned radioligands have yet been fully validated for clinical purposes. In a recent study, Biogen Inc unveiled BIO-2008846 (BIIB129) [21], a novel brain penetrant targeted covalent inhibitor (TCI) of BTK, characterized by its unique structure and binding mode that confers high kinome selectivity. BIO-2008846 has been shown to suppress whole blood CD69 activation and cellular proliferation in TMD8 cells, with a maximum effect (Emax) of 100% and IC_50_ values WB CD69 IC_50_ = 0.079 μM, TMD8 IC_50_ = 0.82 nM. BIO-2008846 showcased promising efficacy in relevant preclinical in vivo models targeting B-cell proliferation in the central nervous system (CNS). Moreover, it displayed a favorable safety profile conducive to clinical advancement as an immunomodulating therapy for multiple sclerosis (MS) with a projected low total human daily dose.

In this work, our aim was (a) to perform radiolabelling of BIO-2008846 at three different positions with carbon-11, (b) to conduct PET measurements in NHP using BIO-2008846-A to assess brain kinetics, and finally, (c) to analyze radiometabolites in NHP blood plasma.

## 2. Results and Discussions

### Radiochemistry

The radiolabeling process for [^11^C]BIO-2008846-A and [^11^C]BIO-2008846-B involved a one-pot [^11^C]methylation reaction using desmethyl precursors PRE-1 and PRE-2 (Figure 1). The cyclotron target produced [^11^C]-CH_4_, which was converted to ^11^C-CH_3_I, the alkylating agent necessary for the [^11^C]methylation reaction. The entire radiosynthesis, purification, and formulation process was completed in approximately 30 min. Implementing N-alkylation as shown in Figure 1 and utilizing [^11^C]CH_3_I, the process demonstrated high reproducibility, resulting in the production of over 1000 MBq of the final pure product after irradiating the cyclotron target with a 35 μA beam current for 15–20 min. The molar activity (MA) of [^11^C]BIO-2008846-A ranged from 359 to 552 GBq/µmol at the time of injection into non-human primates (NHP), while the MA of [^11^C]BIO-2008846-B exceeded 800 GBq/µmol by the end of the synthesis. Upon completion of the synthesis, the radiochemical purity surpassed 99%, confirmed by co-injection with an authentic reference standard using HPLC equipped with UV and radio detectors. The final product, [^11^C]BIO-2008846-A or [^11^C]BIO-2008846-B, was formulated in a sterile saline solution and remained stable with a radiochemical purity of over 97% for up to 60 min.

The synthesis of [^11^C]BIO-2008846-A and [^11^C]BIO-2008846-B was carried out using the fully automated synthesizer (Scansys PET chemistry module) provided by Scansys Laboratorieteknik in Denmark. To enhance the reaction efficiency, a range of parameters were meticulously explored. This included testing different reaction solvents such as acetone, methanol, DMF, DMSO, and acetonitrile, as well as utilizing diverse bases like NaOH, KOH, NaH, and K_2_CO_3_. Additionally, the temperature varied from room temperature (RT) up to 100 °C, and adjustments were made to the precursor quantities as outlined in Table 1. In the preliminary stages, alkylating agents—either carbon-11 labeled methyl triflate ([^11^C]CH_3_OTf) or methyl iodide ([^11^C]CH_3_I)—were employed. Noteworthy progress was achieved when a combination of precursors in the range of 0.6–1.0 mg and [^11^C]CH_3_I in DMSO at 100 °C was used, resulting in the successful production of the desired product with a considerably improved radiochemical yield (RCY) exceeding 25%. With this optimized method, the final product yield ranged between 900 and 1500 MBq, a quantity deemed adequate for subsequent in vivo evaluations.

In the production process of [^11^C]BIO-2008846-C, [^11^C]CO_2_ generated from the cyclotron target was employed to create [^11^C]CO, which served as the radiolabeling agent. This involved a fully automated procedure for the production of [^11^C]CO, followed by the trapping and integration of the generated [^11^C]CO into the precursor. These steps were executed in accordance with established methodologies that have been previously documented [22]. To optimize the reaction, a thorough exploration was conducted, considering various reaction solvents such as DMF, DMSO, tetrahydrofuran, and acetonitrile, along with different temperature settings ranging from room temperature to 120 °C. The best outcome was attained by employing Pd(dba)2, NiXantphos, and a precursor amine (2.0 mg) in THF as the solvent, while maintaining a temperature of 90 °C for 6 min.

The entire radiosynthesis process, encompassing HPLC purification, SPE isolation, and the formulation of [^11^C]BIO-2008846-C, was completed in approximately 45 min. A one-step ^11^C-acylation process utilizing [^11^C]CO resulted in the production of 600–850 MBq of the pure final product. This was achieved following irradiation of the cyclotron target with a beam current of 35 μA for 15 min. The molar activity of the radioligand produced was estimated to be around 30 GBq/µmol by the conclusion of the synthesis. Moreover, the radiochemical purity of the product at the end of the synthesis exceeded 99%. Confirmation of the identity of the radioligand was ensured through co-injection with an authentic reference standard using HPLC equipped with both UV and radio detectors. When formulated in sterile saline, the final product [^11^C]BIO-2008846-C exhibited remarkable stability, maintaining a radiochemical purity exceeding 99% for up to 60 min. Despite the successful labeling of [^11^C]BIO-2008846 at three different positions, we opted to proceed with the biological evaluation using [^11^C]BIO-2008846-A. This radioligand demonstrated superior reproducibility and higher yield compared to the other two options. Two male cynomolgus non-human primates (NHP1 and NHP2) were analyzed using [^11^C]BIO-2008846-A (as detailed in Table 1). At the time of injection, the radioactivity of [^11^C]BIO-2008846-A was measured at 149 ± 20 MBq with an injected mass of 0.12 ± 0.04 µg. Each NHP underwent two PET measurements on the same day, starting with the baseline condition following the first PET scan after the administration of [^11^C]BIO-2008846-A and the second PET scan after pretreatment with non-radiolabeled BIO-2008846. Averaged PET images for both the two baseline and the two blocking studies, along with T1w MRI for anatomical reference, are visible in Figure 1A,B.

The peak whole brain uptakes of [^11^C]BIO-2008846-A during baseline conditions were 1.9 and 2.7 standardized uptake values (SUV) for NHP1 and NHP2, respectively (Figure 2A,B). Initially, there was a rapid and uniform increase in radioligand uptake observed throughout the brain, maintaining consistent SUV levels across various brain regions (Figure 2A,B). The tracer demonstrated rapid washout in all brain regions, indicating reversible kinetics (depicted in Figure 3A,B and Figure 4A,B). TACs for regional radioactivity from all four measurements involving arterial blood sampling were assessed using kinetic modeling using a two-tissue compartment model (2TCM) or graphical analyses using Ichise Multilinear Analysis MA1 (MA1) providing total distribution volume (V_T_) [23] (the results have been detailed in the Appendix A for reference). Preference was given to the MA1 model over the 2TCM model for most measurements and brain regions because the latter failed to properly fit in several regions. A substantial increase in V_T_, estimated using MA1, compared to baseline was observed after pretreatment with BIO-2008846 (refer to Table 2). This increase in uptake and V_T_ could partly be explained by an increase in plasma free fraction (~10%), but not fully since the increase was about 50–100%. The results indicate a lack of specific binding of [^11^C]BIO-2008846-A to BTK, providing uncertainty in the modeling and further analyses of pretreatment PET data. Due to the increase in V_T_ rather than a decrease, analyses of target occupancy using a revised Lassen plot were not possible.

Following deproteinization, more than 95% of the radioactivity present in plasma was efficiently recovered into acetonitrile. HPLC analysis post-injection of [^11^C]BIO-2008846-A in plasma indicated the compound’s elution from the column at a retention time of 5.3 min. Initially, the parent compound was most prevalent at 2 min, constituting roughly 80% and decreasing to around 10% at 30 min for PET studies, as depicted in the percentage of unchanged radioligand in plasma across four PET studies (Figure 5). Subsequently, several more polar radiometabolite peaks were visible, eluted before the parent peak (Figure 2). Verification of [^11^C]BIO-2008846-A’s identity was confirmed through co-injection with non-radioactive BIO-2008846-A. Plasma protein binding assessments were carried out during the four PET sessions, and the corresponding results of [^11^C]BIO-2008846-A for each NHP are outlined in Table 3. In evaluating the binding of [^11^C]BIO-2008846-A to NHP plasma proteins, the ultrafiltration method was utilized [24]. The outcomes were adjusted based on control samples to accommodate membrane binding considerations.

The study’s investigation into the brain uptake of the [^11^C]BIO-2008846 radioligand has produced promising results. Despite observing slight variability in distribution across different brain regions, the kinetics suggest a pattern characteristic of passive diffusion, with free concentration and non-specific binding playing prominent roles in the process. Notably, healthy brain tissue exhibits low levels of BTK expression, primarily confined to microglial cells, representing a small proportion (5–10%) of total CNS cells. In disease contexts like multiple sclerosis (MS), the potential migration of BTK-expressing B cells to the brain and subsequent formation of follicles around the brain parenchyma is acknowledged [25]. However, it is projected that the cell count within these follicles will be minimal. Additionally, signals originating from the bloodstream and circulating B cells expressing BTK contribute to the overall PET signal within the region of interest. As a collective outcome, it is foreseen that specific binding will not be detectable under these circumstances.

## 3. Materials and Methods

### 3.1. Radiochemistry

#### General

Biogen MA Inc. 225 Binney St., Cambridge, MA, USA supplied all the precursors (N-((1s,3s)-3-methyl-3-((6-(1-methyl-1H-pyrazol-4-yl)pyrazolo [1,5-a]pyrazin-4-yl)oxy)cyclobutyl) acrylamide) (PRE-1), (N-((1s,3s)-3-((6-(1H-pyrazol-4-yl)pyrazolo [1,5-a] pyrazin-4-yl)oxy)-3-methylcyclobutyl)-N-methylacrylamide) (PRE-2), and ((1s,3s)-N,3-dimethyl-3-((6-(1-methyl-1H-pyrazol-4-yl)pyrazolo [1,5-a]pyrazin-4-yl)oxy)cyclobutan-1-amine) (PRE-3). They also provided the reference standard BIO-2008846 (N-methyl-N-((1s,3s)-3-methyl-3-((6-(1-methyl-1H-pyrazol-4-yl)pyrazolo [1,5-a]pyrazin-4-yl)oxy)cyclobutyl)acryl amide). The characterization data of precursors and reference compounds are available in Appendix A. We obtained all the other chemicals and reagents from commercial sources and used them without any additional purification. To perform solid-phase extraction (SPE), we bought SepPak C18 Plus cartridges from Waters in Milford, MA, USA. The C-18 Plus cartridge was prepared by activating it first with 10 mL of EtOH and then with 10 mL of sterile water. For liquid chromatographic analysis (LC), we used a Merck-Hitachi gradient pump and a Merck-Hitachi L-4000 variable wavelength UV detector.

### 3.2. Synthesis of ^11^C-Methyliodide (^11^C-CH_3_I)

The synthesis of [^11^C]methyliodide ([^11^C]CH_3_I) is a complex process that involves several steps. It follows a previously published method with some minor modifications [26]. To start, [^11^C]methane ([^11^C]CH_4_) is generated in a GEMS PET trace cyclotron using 16.4 MeV protons. This reaction occurs between nitrogen and 10% hydrogen and requires irradiation of the target gas for approximately 15–20 min with a beam current of 35 μA. Once [^11^C]CH_4_ is produced, it is collected in a Porapak Q trap that is cooled using liquid nitrogen. This trap allows for the isolation and preservation of [^11^C]CH_4_. The next step involves transforming [^11^C]CH_4_ into [^11^C]CH_3_I. The trapped [^11^C]CH_4_ is heated and mixed with vapors from iodine crystals at a temperature of 60 °C. This mixture undergoes a radical reaction within a closed circulation system, which takes place at a high temperature of 720 °C. The resulting [^11^C]CH_3_I is then collected in another Porapak Q trap at room temperature. Any unreacted [^11^C]CH_3_I is recirculated for 3 min to ensure the completion of the reaction and maximize the yield. Finally, the [^11^C]CH_3_I is released from the trap by heating it to 180 °C while allowing a flow of helium. This process enables the retrieval of the synthesized [^11^C]CH_3_I for further use or analysis.

### 3.3. Synthesis of N-methyl-N-((1s,3s)-3-methyl-3-((6-(1-[^11^C]methyl-1H-pyrazol-4-yl)pyrazolo [1,5-a]pyrazin-4-yl)oxy)cyclobutyl) acrylamide ([^11^C]BIO-2008846-A) and N-[^11^C]methyl-N-((1s,3s)-3-methyl-3-((6-(1-methyl-1H-pyrazol-4-yl)pyrazolo [1,5-a]pyrazin-4-yl)oxy)cyclobutyl) acrylamide ([^11^C]BIO-2008846-B)

To obtain [^11^C]-labeled compounds ([^11^C]BIO-2008846-A and [^11^C]BIO-2008846-B), the process involved trapping [^11^C]CH_3_I at room temperature within a reaction vessel containing an appropriate mixture of precursor (PRE1 or PRE2), base, and solvents. Once trapping was complete, the reaction mixture was then heated at ambient temperature. Prior to injection into the integrated high-performance liquid chromatography (HPLC) system for purification of the radiolabeled compound, the reaction mixture was diluted with sterile water (500 µL). The HPLC system used consisted of a semi-preparative reverse phase (RP) ACE column (C18, 10 × 250 mm, 5 µm particle size) and a Merck Hitachi UV detector (λ = 254 nm) (VWR, International, Stockholm, Sweden), connected in series with a GM-tube (Carroll-Ramsey, Berkley, CA, USA) for radioactivity detection. The product was eluted using a mobile phase comprising 40% acetonitrile in trifluoroacetic acid (TFA, 0.1%), with a flow rate of 6 mL/min. This resulted in a radioactive fraction corresponding to the pure tracer, with a retention time (tR) of 9 min.

### 3.4. Synthesis of [^11^C]Carbon Monoxide ([^11^C]CO)

The synthesis of [^11^C]carbon monoxide ([^11^C]CO) followed a previously published method [22]. Initially, [^11^C]CO_2_ without any carrier was generated by subjecting a mixture of nitrogen and oxygen gas (0.5% oxygen) to a 14N(p,α)11C nuclear reaction using 16.5 MeV protons produced by the GEMS PET trace cyclotron (GE, Uppsala, Sweden). At the end of bombardment (EOB), the target content was transferred to the [^11^C]CO synthesizer prototype. Here, the [^11^C]CO_2_ was captured on a silica gel trap (10 mg, 60 Å, 60–100 mesh) immersed in liquid nitrogen. The concentrated [^11^C]CO_2_ was subsequently released from the trap through thermal heating. To convert it to [^11^C]CO, the [^11^C]CO_2_ was introduced into a pre-heated quartz glass column (6 × 4 × 180 mm: outer diameter × inner diameter × length) filled with molybdenum powder (1.5 g, <150 µm, 99.99% trace metals basis, Sigma Aldrich) for reduction. The resulting [^11^C]CO was then trapped and concentrated on another silica gel trap (10 mg, 60 Å, 60–100 mesh) immersed in liquid nitrogen. Any remaining unreacted [^11^C]CO_2_ was subsequently removed using a sodium hydroxide-coated silica trap (0.2 g, Ascarite II, 20–30 mesh) placed inside a 30 mm 1/8” stainless steel tube. After completing the trapping process, the trap was heated through thermal heating to release the [^11^C]CO for further use.

### 3.5. Synthesis of N-methyl-N-((1s,3s)-3-Methyl-3-((6-(1-Methyl-1H-Myrazol-4-yl)Pyrazolo [1,5-a]Pyrazin-4-yl)oxy)cyclobutyl)([^11^C]CO)acrylamide ([^11^C]BIO-2008846-C)

[^11^C]BIO-2008846-C was synthesized by trapping [^11^C]CO at room temperature within a reaction vessel containing a mixture of the amine precursor (PRE3, 2 mg, 0.006 mmol), palladium(II) complex (Pd(dba)_2_, 3.5 mg), and N-XantPhos (3 mg) in THF (400 µL). The reaction mixture was then heated at 150 °C for 300 s. The resulting residue was diluted with sterile water (2 mL) and injected into the HPLC injection loop for purification. The injection loop was connected to the integrated high-performance liquid chromatography (HPLC) system, equipped with a semi-preparative reverse phase (RP) ACE column (C18, 10 × 250 mm, 5 µm particle size). The column outlet was connected to a Merck Hitachi UV detector (λ = 254 nm) (VWR, International, Stockholm, Sweden) in series with a GM-tube (Carroll-Ramsey, Berkley, CA, USA) for radioactivity detection. For the HPLC purification, an isocratic mobile phase consisting of a mixture of acetonitrile (35%) and TFA (0.1%) (85%) was used, with a flow rate of 6 mL/min. The radioactive fraction corresponding to the desired product [^11^C]BIO-2008846-C was eluted with a retention time (t_R_) of 900 s and then diluted with sterile water (50 mL). The resulting mixture was passed through a conditioned SepPak tC18 plus cartridge. The cartridge was washed with sterile water (10 mL), and the isolated [^11^C]BIO-2008846-C was subsequently eluted with 1 mL of ethanol into a sterile vial containing sterile saline (9 mL). The formulated product was then sterile filtered using a Millipore Millex^®^ GV filter unit (0.22 μm) for further use.

### 3.6. Study Design in Non-Human Primates (NHP), PET Procedure and Quantification

This research study followed ethical guidelines and received approval from the Animal Ethics Committee in Stockholm of the Swedish Board of Agriculture (10367-2019). The study also adhered to the guidelines outlined in “Guidelines for planning, conducting and documenting experimental research” (Dnr 4820/06-600) of the Karolinska Institutet. The non-human primates (NHPs) used in this study were housed in the Astrid Fagraeus Laboratory, Comparative Medicine in Solna, Sweden. Two male cynomolgus NHPs were chosen to participate in the study. The study involved conducting brain PET scans on both NHPs under baseline conditions. Afterward, a pretreatment of BIO-2008846 at a dose of 1.0 mg/kg body weight was given intravenously as a bolus over about 5 min, approximately 7 min prior to administering the radioligand [^11^C]BIO-2008846-A. The pretreatment formulation consisted of BIO-2008846 at a concentration of 1.0 mg/mL in a solution of 3% DMSO and 5% Kolliphor HS15 in saline (15 mL). Arterial blood samples were collected to measure the plasma input function.

The anesthesia for the NHPs was induced through intramuscular injection of ketamine hydrochloride (10 mg/kg) at the Astrid Fagraeus Laboratory and maintained using a mixture of sevoflurane, oxygen, and medical air delivered through endotracheal intubation. A fixation device was used to immobilize the head, and body temperature was monitored and maintained using a Bair Hugger model 505 warming unit (Arizant Healthcare, Eden Prairie, MN, USA) along with an esophageal thermometer. Throughout the experiments, continuous monitoring of heart rate, blood pressure, respiratory rate, and oxygen saturation was carried out. The NHPs were also kept in fluid balance through continuous saline infusion. PET measurements were performed using a High-Resolution Research Tomograph (HRRT) from (Siemens Molecular Imaging) [27]. Prior to the injection of [^11^C]BIO-2008846-A, a 6-min transmission scan was conducted using a single 137Cs source. Following the intravenous injection of [^11^C]BIO-2008846-A, list mode data were continuously acquired for a duration of 123 min. Afterward, the acquired images underwent reconstruction using the ordinary Poisson-3D-ordered subset expectation maximization (OP-3D-OSEM) algorithm. The reconstruction process consisted of 10 iterations and 16 subsets, and it also incorporated modeling of the point spread function (PSF).

To obtain brain magnetic resonance imaging (MRI) data, a 3.0T DISCOVERY MR750 from General Electric (Milwaukee, WI, USA) was utilized. A T1-weighted image was acquired for the purpose of co-registering with PET scans and delineating anatomic brain regions.

To analyze the data, regions of interest (ROIs) were manually delineated on MRI images of each NHP, specifically for the whole brain, cerebellum, caudate, putamen, thalamus, frontal cortex, occipital cortex, and hippocampus. The PET images capturing the entire duration were then co-registered to the individual NHP’s MRI image. The co-registration parameters were applied to the dynamic PET data, allowing for the generation of time–activity curves for the different brain regions for each PET measurement. The average standardized uptake value (SUV) was calculated for each brain region.

### 3.7. Radiometabolite Analysis and Protein Binding in Plasma

The radiometabolites in the blood plasma of non-human primates (NHP) were analyzed using a method that had been previously published [24]. During the entire PET measurement process, the percentages of radioactivity corresponding to [^11^C]BIO-2008846-A and its radioactive metabolites in the blood plasma of the NHP were determined using a reverse-phase high-performance liquid chromatography (HPLC) system. The HPLC system utilized a UV absorbance detector with a wavelength of 254 nm coupled with a radioactive detector. Arterial blood samples, ranging from 0.7 to 1.5 mL, were collected manually at various time intervals such as 1, 1.5, 2, 2.5, 3, 5, 15, 30, 45, 60, 90 and 120 after administering [^11^C]BIO-2008846-A. Radiometabolites were analyzed in blood samples from the following time points: 2, 5, 15, 30, 60, 75, and 90 min. After collection, the blood was centrifuged at 4000 rpm for 2 min to separate the plasma. The plasma was then diluted with 1.4 times its volume of acetonitrile and centrifuged at 6000 rpm for 4 min. The resulting extract was separated from the pellet and further diluted with water (3 mL). For the chromatographic separation of radiometabolites from unchanged [^11^C]BIO-2008846-A, a high-performance liquid chromatography (HPLC) system was utilized. The HPLC system consisted of an Agilent binary pump (Agilent 1200 series) connected to a manual injection valve (Rheodyne 7725i), a 5.0 mL loop, and a radiation detector (Oyokoken S-2493Z) housed within a 50 mm lead shield. A semi-preparative reverse phase ACE 5µm C18 HL column (250 × 10 mm) was used, employing gradient elution to achieve the desired chromatographic separation. The mobile phase comprised acetonitrile (A) and 0.1 M ammonium formate (B) at a flow rate of 5.0 mL/min. The following program was followed: 0–4.0 min, (A/B) 40:60 *v*/*v* to 90:10 *v*/*v*; 4.0–6.0 min, (A/B) 90:10 *v*/*v*; 6.0−6.1 min, (A/B) 90:10 to 40:60 *v*/*v*. The radioactive peak corresponding to [^11^C]BIO-2008846-A was integrated, and its area was expressed as a percentage of the total areas of all detected radioactive compounds. To determine the recovery of radioactivity from the system, a 2 mL aliquot of the eluate from the HPLC column was measured and divided by the total injected radioactivity. 

Before the injection of [^11^C]BIO-2008846-A, the free fraction (fp) in blood plasma was measured using an ultrafiltration method [24]. In this method, 0.4 mL of plasma was mixed with 0.04 mL of [^11^C]BIO-2008846- The sample (~1 MBq) was formulated and then left to incubate at room temperature for 10 min. To assess the non-specific binding percentage, the same procedure was carried out using phosphate-buffered saline (PBS) (0.4 mL) as a control solution. After the incubation period, 0.2 mL of the plasma and PBS mixtures were transferred into Centrifree YM-30 ultrafiltration tubes with a molecular weight cutoff of 30,000 Da (Millipore: Billerica, MA, USA) and centrifuged at 3800 rpm for 15 min. Following centrifugation, equal volumes (0.02 mL) of the resulting ultrafiltrate (Cfree) and the plasma (Ctotal) were analyzed for radioactivity using a NaI well counter. Each measurement was repeated in triplicate. The free fraction (fp) was then determined by dividing Cfree by Ctotal. The results were adjusted to compensate for any membrane binding effects by using the control samples.

HPLC for plasma protein binding was performed using a radio-HPLC system consisting of an interface module (D-7000; Hitachi: Tokyo, Japan), a L-7100 pump (Hitachi), an injector (model 7125, with a 5.0-mL loop; Rheodyne: Cotati, CA, USA), and an ultraviolet absorption detector (L-7400, 254 nm; Hitachi) in series with a 150TR Packard radioactivity detector (housed in a shield of 50 mm thick lead) equipped with a 550 μL flow cell. Chromatographic separation was achieved on an ACE 5µm C18 HL column (250 × 100 mm). A gradient elution of acetonitrile (A) and ammonium formate (AMF) 0.1 M (B) was used as the mobile phase at 6.0 mL/min, with the following gradient: 0–4.0 min, (A/B) 40:60 *v*/*v* to 90:10 *v*/*v*; 4.0–6.0 min, (A/B) 90:10 *v*/*v*; 6.0−6.1 min, (A/B) 90:10 to 40:60 *v*/*v*. 

## 4. Conclusions

The study effectively demonstrated the successful labeling of the radioligand BIO-2008846 with carbon-11 at three different positions. PET scans conducted on cynomolgus NHPs revealed a significant uptake of the radioligand [^11^C]BIO-2008846-A in the brain. Interestingly, no blocking effect was observed following pretreatment with non-radioactive BIO-2008846; instead, an increase in uptake and slower washout were noted. These results suggest that [^11^C]BIO-2008846 might not exhibit strong specific binding to BTK for tracking biodistribution in the brain. Rather, its potential lies in measuring and determining CNS exposure rather than imaging the distribution of BTK.

## Data Availability

All the supporting data are stored at Karolinska Institutet’s archive.

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
