# Peer review of "Labeling of Bruton’s Tyrosine Kinase (BTK) Inhibitor [11C]BIO-2008846 in Three Different Positions and Measurement in NHP Using PET"

_ijms, 2024, doi:10.3390/ijms25147870_

Round 1

Reviewer 1 Report

Comments and Suggestions for Authors

The authors described radiolabeling of [11C]BIO-2008846 using three synthetic routes. [11C]BIO-2008846-A was then evaluated in NHP PET as a potential BTK PET tracer. The work includes advanced technical experiments and provides detailed descriptions of the methodology. A PET tracer for imaging BTK in brain remains of high interest to the field but unfortunately [11C]BIO-2008846 does not appear to provide a breakthrough.

Main comments:

-       The authors appropriately mentioned that several PET tracers have previously been described for imaging of BTK, in particular for oncology application. It is not clear from the introduction if previous tracers could be used for imaging BTK in brain. Please elaborate on the tracer literature in the introduction, as it is currently not clear how much advancement the current work provides.

-          No BTK specific binding in brain was demonstrated in this study. This was not anticipated and should be discussed in more detail. Information should be included detailing the known distribution of BTK receptors in the normal brain (e.g., cell type, regional distribution, Bmax, etc) and disease brain. Are there species differences, rodent vs NHP vs human?

-       There is high expression of BTK on blood cells. Could BTK specific binding of the tracer be demonstrated using blood cells?

-       What is the background of BIO-2008846? Was it tested preclincally or clinically as a therapeutic? Does it have high BTK affinity? What are other properties of BIO-2008846 that make it suitable as a potential PET tracer?

-       The quantification of radioligand binding was poorly described. For example, the abstract refers to the use of a revised Lassen plot for calculating distribution, while MA1 was mentioned in the main body. Please correct the relevant sections and provide more details (e.g., fitting of curves) of the kinetic analysis.

-       Evaluation of the time stability of Vt values should be included and will help evaluate the potential presence of brain penetrant radiolabeled metabolites.

 Minor comments:

Throughout the manuscript there are errors in the naming of the compounds:

-          Line 24 - [11C]BIO-008846-A should read [11C]BIO-2008846-A

-          Table 1: Clarify that this was radioligand A.

-          Line 189, should read BIO-2008846 and not [11C]BIO-2008846

-          What is the difference between compound BBIB104 (mentioned in figure 1 legend) and BIO-2008846?

-          Etc.

Results and discussion:

-          Please provide more details about the pretreatment experiments. What was the route and formulation?

-          Figure 1, please include color bar with scale.

-          Line 156 – is there a word missing between “ing” and “model”?

-          Line 161, it should be specified that free fraction refers to plasma free fraction.

-          Line 162 “specific binding of” is mentioned twice.

Comments on the Quality of English Language

-

Author Response

Reviewer 1

Comments and Suggestions for Authors

The authors described radiolabeling of [11C]BIO-2008846 using three synthetic routes. [11C]BIO-2008846-A was then evaluated in NHP PET as a potential BTK PET tracer. The work includes advanced technical experiments and provides detailed descriptions of the methodology. A PET tracer for imaging BTK in brain remains of high interest to the field but unfortunately [11C]BIO-2008846 does not appear to provide a breakthrough.

Main comments:

The authors appropriately mentioned that several PET tracers have previously been described for imaging of BTK, in particular for oncology application. It is not clear from the introduction if previous tracers could be used for imaging BTK in brain. Please elaborate on the tracer literature in the introduction, as it is currently not clear how much advancement the current work provides.

No BTK specific binding in brain was demonstrated in this study. This was not anticipated and should be discussed in more detail. Information should be included detailing the known distribution of BTK receptors in the normal brain (e.g., cell type, regional distribution, Bmax, etc) and disease brain. Are there species differences, rodent vs NHP vs human?

Reply: Amidst the ongoing research on BTK PET tracers, it is notable that the majority of these tracers ([11C]tolebrutinib, [11C]ibrutinib, [11C]evobrutinib, and various 18F-labeled PET tracers like [18F]BTK-1, [18F]ABBV-BTK1, and [18F]PTBTK3) are currently in the preclinical phase. This indicates that they are undergoing evaluation in animal models to determine their safety, effectiveness, and suitability for potential human application. As of now, none of these tracers have advanced to full-scale clinical trials involving human patients.

The aim of this study was not to study specific binding to BTK since the expression of the target in healthy brain is extremely low (https://pubmed.ncbi.nlm.nih.gov/38022591/). . We rather aimed to evaluate the brain uptake and kinetics of a molecule developed for treating pathology in the CNS (MS) with the intention to evaluate brain exposure and de-risk the clinical program.

A novel brain-penetrant targeted covalent inhibitor (TCI) of BTK, known as BIO-2008846 (BIIB129) (Ref. 21), was recently introduced by Biogen Inc in a study. Distinguished by its distinctive structure and binding mode, BIO-2008846 exhibits high selectivity within the kinome. In preclinical in vivo models, BIO-2008846 demonstrated promising efficacy in targeting B cell proliferation within the central nervous system (CNS). Additionally, it showed a favorable safety profile that supports its progression into clinical development as a potential immunomodulating therapy for multiple sclerosis (MS), with an anticipated low total daily dose for human use.

There is high expression of BTK on blood cells. Could BTK specific binding of the tracer be demonstrated using blood cells?

Reply: It is a valid point that binding to blood cells could have an impact on the signal seen in brain. The kinetic modelling with 2TCM did not reveal any unusual contribution from blood compartment and the model provided a fit with an estimated blood contribution about the predicted 5%. We used arterial blood sampling and full kinetic modelling to include radioligand binding to blood components in the evaluation of Distribution volume in brain, thus accounting for changes in the metabolite corrected input function due to radioligand binding to blood cells.

What is the background of BIO-2008846? Was it tested preclinically or clinically as a therapeutic? Does it have high BTK affinity? What are other properties of BIO-2008846 that make it suitable as a potential PET tracer?

Reply: In a recent study, Biogen Inc unveiled BIO-2008846 (BIIB129), a novel brain-penetrant targeted covalent inhibitor (TCI) of BTK, characterized by its unique structure and binding mode that confers high kinome selectivity. BIO-2008846 showcased promising efficacy in relevant preclinical in vivo models targeting B cell proliferation in the central nervous system (CNS). Moreover, it displayed a favorable safety profile conducive to clinical advancement as an immunomodulating therapy for multiple sclerosis (MS) with a projected low total human daily dose (Discovery and Preclinical Characterization of BIIB129, a Covalent, Selective, and Brain-Penetrant BTK Inhibitor for the Treatment of Multiple Sclerosis. J Med Chem 2024, 67 (10), 8122-8140). This text is added in the introduction section.

The quantification of radioligand binding was poorly described. For example, the abstract refers to the use of a revised Lassen plot for calculating distribution, while MA1 was mentioned in the main body. Please correct the relevant sections and provide more details (e.g., fitting of curves) of the kinetic analysis.

Reply: Thank you for the suggestion. Since the aim with the study was to evaluate a novel drug and not a new radioligand the level of detail in the quantification is lower than if we would have had another aim. The goal to provide quantifiable out come measures that could be compared between baseline and post-dose PET, not to evaluate the kinetic modelling per se. We have also changed the text in the abstract to give relevant information on quantification of VT and not estimation of Occupancy using Lassen plot.

Evaluation of the time stability of Vt values should be included and will help evaluate the potential presence of brain penetrant radiolabeled metabolites.

Reply:We did not on the radiometabolite analyses find any radio-peaks from metabolites more lipophilic than the parent compound. Together with the fact that the shape of the time-activity-curves showed washout throughout the experiment we did not find evidence supporting the presence of radiolabeled metabolites. Given the fact that the aim was to evaluate a novel drug and not a radioligand we do not think that elaborate time-stability analyses are of key interest given the focus of the project.

 Minor comments:

Throughout the manuscript there are errors in the naming of the compounds:

-          Line 24 - [11C]BIO-008846-A should read [11C]BIO-2008846-A

Reply: Corrected accordingly.

-          Table 1: Clarify that this was radioligand A.

Reply: Corrected accordingly.

-          Line 189, should read BIO-2008846 and not [11C]BIO-2008846

Reply: Corrected accordingly.

-          What is the difference between compound BBIB104 (mentioned in figure 1 legend) and BIO-2008846?

      - BIIB104 is just another name for BIO-2008846, right?

Reply: Typing error, corrected accordingly.

Results and discussion:

Please provide more details about the pretreatment experiments. What was the route and formulation?

Reply: A pretreatment of BIO-2008846 at a dose of 1.0 mg/kg body weight was given intravenously as a bolus over about 5 minutes, approximately 7 minutes prior to administering the radioligand [11C]BIO-2008846-A. The pretreatment formulation consisted of BIO-2008846 at a concentration of 1.0 mg/ml in a solution of 3% DMSO and 5% Kolliphor HS15 in saline (15 mL). Text is added in the experimental section.

-          Figure 1, please include color bar with scale.

Reply: Corrected accordingly.

-          Line 156 – is there a word missing between “ing” and “model”?

Reply: Corrected accordingly.

-          Line 161, it should be specified that free fraction refers to plasma free fraction.

Reply: Corrected accordingly.

-          Line 162 “specific binding of” is mentioned twice.

Reply: Corrected accordingly.

Reviewer 2 Report

Comments and Suggestions for Authors

The article titled " Labelling of a Bruton’s Tyrosine Kinase (BTK) Inhibitor [11C]BIO-2008846 in Three Different Position and Measurement in NHP Using PET" performed synthesis and evaluation of PET tracers for the visualization of BTK kinase in NHPs brain. [11C]BIO-2008846-A can cross the blood-brain barrier and exhibits non-specific binding in brain. The work could be acceptable for this journal after minor revision. The following are the questions in this manuscript:

Comment 1: A short and logical Abstract is needed in this manuscript.

Comment 2: More pharmacology and physiochemical properties of three reference compounds (BIO-2008846-A, BIO-2008846-B, and BIO-2008846-C) need to be discussed, including the binding affinity with BTK kinase, LogD, and cLogP.

Comment 3: BTK kinase plays an important role in blood tumors, why did the authors evaluate the distribution in the brain instead of tumors? Please give more explanations in your manuscript.

Comment 4: Please provide the characterization data of precursors and reference compounds, including LC/MS and H-NMR.

Author Response

The article titled " Labelling of a Bruton’s Tyrosine Kinase (BTK) Inhibitor [11C]BIO[1]2008846 in Three Different Position and Measurement in NHP Using PET" performed synthesis and evaluation of PET tracers for the visualization of BTK kinase in NHPs brain. [11C]BIO-2008846-A can cross the blood-brain barrier and exhibits non-specific binding in brain. The work could be acceptable for this journal after minor revision.

The following are the questions in this manuscript:

Comment 1: A short and logical Abstract is needed in this manuscript.

Reply: Agree with the reviewer and a new updated abstract is added.

Bruton's tyrosine kinase (BTK) is pivotal in B-cell signaling and a target for potential anti-cancer and immunological disorder therapies. Improved selective reversible BTK inhibitors are in demand due to the absence of direct BTK engagement measurement tools. Promisingly, PET imaging can non-invasively evaluate BTK expression. In this study, radiolabeled BIO-2008846 ([11C]BIO-2008846-A), a BTK inhibitor, was used for PET imaging in NHPs with to track brain biodistribution. Radiolabeling BIO-2008846 with carbon-11, alongside four PET scans on two NHPs each, showed homogeneous distribution of [11C]BIO-2008846-A in NHP brains. Brain uptake ranged from 1.8% ID at baseline to a maximum of 3.2% post-pretreatment. The study found no significant decrease in regional VT values post-dose, implying minimal specific binding of [11C]BIO-2008846-A compared to free and non-specific components in brain. Radiometabolite analysis revealed polar metabolites with 10% unchanged radioligand after 30 minutes. The research highlighted strong brain uptake despite minor distribution variability, confirming passive diffusion kinetics dominated by free and non-specific binding.

Comment 2: More pharmacology and physiochemical properties of three reference compounds (BIO-2008846-A, BIO-2008846-B, and BIO-2008846-C) need to be discussed, including the binding affinity with BTK kinase, LogD, and cLogP.

Reply: New text including reference is added in the manuscript.

A recent study by Biogen Inc introduced BIO-2008846 (BIIB129), a groundbreaking brain-penetrant targeted covalent inhibitor (TCI) of BTK, distinguished by its exceptional structure, pharmacology, physiochemical properties, and binding mode that grants it high kinome selectivity. In the study, BIO-2008846 exhibited promising efficacy, displaying considerable binding affinity with BTK kinase, LogD, and cLogP in pertinent preclinical in vivo models aimed at addressing B cell proliferation within the central nervous system (CNS). (Discovery and Preclinical Characterization of BIIB129, a Covalent, Selective, and Brain-Penetrant BTK Inhibitor for the Treatment of Multiple Sclerosis. J Med Chem 2024, 67 (10), 8122-8140).

Comment 3: BTK kinase plays an important role in blood tumors, why did the authors evaluate the distribution in the brain instead of tumors? Please give more explanations in your manuscript.

Reply: We do not have to possibility to do PET in NHPs with any diseases, only healthy. The aim of this study was not to specifically study binding since the expression of BTK in healthy brain is not as high as in specific types of cancer or other diseases such as MS or RA. The aim was to evaluate the brain uptake and kinetics to be further evaluated in disease models and further on in patient populations where there would be changes in levels of BTK.

Comment 4: Please provide the characterization data of precursors and reference

compounds, including LC/MS and H-NMR.

Reply: Added as supporting material

Round 2

Reviewer 1 Report

Comments and Suggestions for Authors

Figure 1:           BIIB104, an AMPA PAM, is mentioned in Figure 1 legend. This must be a remaining typo.

Page 2:             It would be helpful if authors could report the BTK affinity of BIO-2008846.

Page 7:             The sentence “TACs for regional radioactivity from all four measurements involving arterial blood sampling were assessed using kinetic modelling using two-tissue compartment model (2TCM) or graphical analyses us-ing  model and Ichise Multilinear Analysis MA1 (MA1) providing total distribution volume ratio (VT)[23].” remains confusing. Is there a word missing between “us-ing” and “model”? Also, “distribution volume ratio (VT)” should read “distribution volume (VT)”.

Page 7              It is unclear how well the MA1 model describes the TACs and how reliable the estimate of Vt is. The authors should provide fits of the data and report t* values. It is important for the reader to understand the reliability of the Vt estimates as this is the main outcome measure of the PET study.

Comments on the Quality of English Language

-

Author Response

Reviewer 1

Figure 1:           BIIB104, is mentioned in Figure 1 legend. This must be a remaining typo.

Reply: Corrected accordingly.

Page 2:             It would be helpful if authors could report the BTK affinity of BIO-2008846.

Reply: The following sentence is added in the introduction section. “BIO-2008846 has been shown to suppress whole blood CD69 activation and cellular pro-liferation in TMD8 cells, with a maximum effect (Emax) of 100% and IC50 values WB CD69 IC50 = 0.079 μM, TMD8 IC50 = 0.82 nM”. The details CNS MPO Scores and Physicochemical Properties including BTK affinity of BIO-2008846(BIIB129) has already been reported and published (reference 21).

Page 7:                The sentence “TACs for regional radioactivity from all four measurements involving arterial blood sampling were assessed using kinetic modelling using two-tissue compartment model (2TCM) or graphical analyses us-ing  model and Ichise Multilinear Analysis MA1 (MA1) providing total distribution volume ratio (VT)[23].” remains confusing. Is there a word missing between “us-ing” and “model”? Also, “distribution volume ratio (VT)” should read “distribution volume (VT)”.

Reply: The sentence is rephrased “TACs for regional radioactivity from all four measurements involving arterial blood sampling were assessed using kinetic modelling using two-tissue compartment model (2TCM) or graphical analyses using Ichise Multilinear Analysis MA1 (MA1) providing total distribution volume (VT)”

Page 7:                It is unclear how well the MA1 model describes the TACs and how reliable the estimate of Vt is. The authors should provide fits of the data and report t* values. It is important for the reader to understand the reliability of the Vt estimates as this is the main outcome measure of the PET study.

Reply: Given the striking similarity of the TACs across regions, the resemblance between the 2TCM fits and MA1 plots is quite apparent. The results have been detailed in the supplementary section (S2) for reference.